# Effect of Supplementation with Omega-3 Polyunsaturated Fatty Acids on Metabolic Modulators in Skeletal Muscle of Rats with an Obesogenic High-Fat Diet

**DOI:** 10.3390/ph17020222

**Published:** 2024-02-08

**Authors:** Mara Patricia Chávez-Ortega, Julio Cesar Almanza-Pérez, Fausto Sánchez-Muñoz, Enrique Hong, Elihu Velázquez-Reyes, Rodrigo Romero-Nava, Santiago Villafaña-Rauda, Alfredo Pérez-Ontiveros, Gerardo Blancas-Flores, Fengyang Huang

**Affiliations:** 1Posgrado en Biología Experimental, División de Ciencias Biológicas y de la Salud, Universidad Autónoma Metropolitana, Iztapalapa, Ciudad de México 02200, Mexico; marachavezfth@gmail.com; 2Laboratorio de Investigación en Obesidad y Asma, Hospital Infantil de México Federico Gómez, Ciudad de México 06720, Mexico; alfredonti74@yahoo.com.mx; 3Laboratorio de Farmacología, Departamento de Ciencias de la Salud, División de Ciencias Biológicas y de la Salud, Universidad Autónoma Metropolitana, Iztapalapa, Ciudad de México 02200, Mexico; jcap@xanum.uam.mx (J.C.A.-P.); elihu.velazquez1997@gmail.com (E.V.-R.); 4Departamento de Inmunología, Instituto Nacional de Cardiología Ignacio Chávez, Ciudad de México 14080, Mexico; fausto22@yahoo.com; 5Departamento de Farmacobiología, Centro de Investigación y de Estudios Avanzados del Instituto Politécnico Nacional, Ciudad de México 07360, Mexico; enriquehong@hotmail.com; 6Laboratorio de Señalización Intracelular, Sección de Estudios de Posgrado, Escuela Superior de Medicina del Instituto Politécnico Nacional, Ciudad de México 11340, Mexico; roloromer@gmail.com (R.R.-N.); santiagovr1@gmail.com (S.V.-R.)

**Keywords:** obesity, HFD, omega-3 polyunsaturated fatty acids, PPARs, insulin, skeletal muscle

## Abstract

Previous studies provided evidence of the benefits of omega-3 polyunsaturated fatty acids (ω-3 PUFA) on the cardiovascular system and inflammation. However, its possible effect on skeletal muscle is unknown. This study aimed to evaluate whether ω-3 PUFA reverses the dysregulation of metabolic modulators in the skeletal muscle of rats on a high-fat obesogenic diet. For this purpose, an animal model was developed using male Wistar rats with a high-fat diet (HFD) and subsequently supplemented with ω-3 PUFA. Insulin resistance was assessed, and gene and protein expression of metabolism modulators in skeletal muscle was also calculated using PCR-RT and Western blot. Our results confirmed that in HFD rats, zoometric parameters and insulin resistance were increased compared to SD rats. Furthermore, we demonstrate reduced gene and protein expression of peroxisome proliferator-activated receptors (PPARs) and insulin signaling molecules. After ω-3 PUFA supplementation, we observed that glucose (24.34%), triglycerides (35.78%), and HOMA-IR (40.10%) were reduced, and QUICKI (12.16%) increased compared to HFD rats. Furthermore, in skeletal muscle, we detected increased gene and protein expression of PPAR-α, PPAR-γ, insulin receptor (INSR), insulin receptor substrate 1 (ISR-1), phosphatidylinositol-3-kinase (PI3K), and glucose transporter 4 (GLUT-4). These findings suggest that ω-3 PUFAs decrease insulin resistance of obese skeletal muscle.

## 1. Introduction

Obesity is defined as the excessive accumulation of fatty tissue that can lead to impaired biological functions [1,2]. In 2023, the World Atlas of Obesity estimated that by 2035, approximately 2 billion people will live with obesity [3]. These data are worrying because obesity is a risk factor for developing metabolic disorders [1,4,5].

One of the master modulators of metabolism frequently altered consequently of an obesogenic diet is the peroxisome proliferator-activated receptor (PPAR). Because the expression of the α (PPAR-α) and γ (PPAR-γ) isoforms decreases in the skeletal muscle of obese patients [6,7,8]. The function of the PPAR-α is to prevent lipid storage in muscle by promoting fatty acid oxidation [6,7]. Furthermore, PPAR-γ promotes adiponectin secretion and the expression of relevant genes of the insulin signaling pathway [6,8]. In healthy adults, adiponectin promotes lipid oxidation, insulin sensitivity, and weight loss. However, its secretion decreases consequently in obesity [9,10]. Finally, insulin exerts its effect by binding to the insulin receptor (INSR), which autophosphorylates on tyrosine residues and incorporates phosphate into receptor substrate 1 (ISR-1) [11]. Activation of ISR-1 promotes the formation of molecular complexes of the phosphatidylinositol-3-kinase (PI3K) pathway. Consequently, glucose can enter the cell by translocating glucose transporter 4 (GLUT-4) [11,12]. When this signaling pathway is impeded by insulin resistance, glucose absorption decreases. These alterations in energy metabolism in the muscle play an essential role in the pathogenesis of obesity-related diseases [11].

To reverse the alterations in metabolism modulators, synthetic compounds such as fenofibrate (PPAR-α agonist that lowers triglyceride levels) and pioglitazone (PPAR-γ agonist used to reduce glucose concentration) are recommended [13,14,15]. However, despite their efficacy, fenofibrate and pioglitazone have limitations in their clinical use due to adverse effects [13,15]. For this reason, it is necessary to assess the possible beneficial effect of eicosapentaenoic acid (EPA) and docosahexaenoic acid (DHA), two principal forms of ω-3 PUFA in diet [16].

Studies have provided evidence of the benefits of a diet rich in ω-3 PUFA on the heart, blood vessels, liver, immune system, and endocrine system [16,17]. For this reason, in recent decades, its possible therapeutic effect on cardiovascular diseases, hepatic steatosis, and hypertension has been studied [16]. Furthermore, some publications demonstrated that the beneficial effects of EPA and DHA on metabolic activity are through the activation of PPAR-α and PPAR-γ in adipose tissue and macrophages [16,17,18]. However, the impact of ω-3 PUFA supplementation on these metabolic modulators in skeletal muscle has been little explored. The skeletal muscle plays a central role in regulating an individual’s glucose metabolism since it is primarily responsible for the body’s uptake of glucose [19]. For this reason, the objective of the present study was to evaluate whether ω-3 PUFA supplementation reverts alterations in metabolic modulators associated with a high-fat obesogenic diet in skeletal muscle.

## 2. Results

### 2.1. The Characteristics in the Obese Model Induced by a High-Fat Diet

After 16 weeks of the HFD diet, the animal model of obesity was confirmed with a 23.46% greater weight gain compared to the SD group (Table 1 and Figure 1). Although the water consumption and food intake in the HFD group were lower than in the SD group, the calories consumed were higher (Figure 1). In the case of the zoometric parameters, we obtained an increase in weight, abdominal circumference, and Lee index in the HFD rats compared to the SD rats (Table 1). Finally, the concentrations of triglycerides and glucose in the HFD group were significantly higher than those in the SD group; however, no difference in cholesterol was observed between the groups (Table 1).

### 2.2. Effect of Omega-3 Polyunsaturated Fatty Acids

#### 2.2.1. Zoometric Parameters and Weight of Skeletal Muscle (Soleus)

In the obese rats with ω-3 PUFA, pioglitazone, and fenofibrate, there was no significant difference in body weight, abdominal circumference, and Lee index compared with the HFD group (Table 2). In the case of the skeletal muscle, the relative weight of the soleus muscle in the HFD group was less than in SD rats (Figure 2). However, in the ω-3 PUFA group, no difference was observed in the weight absolute and relative of the soleus muscle.

#### 2.2.2. Biochemical Profile, Adiponectin, and Insulin

In the HFD group, the concentration of glucose and triglycerides increased compared to the SD group. Furthermore, after eight weeks of treatment with ω-3 PUFA, the levels of triglycerides and glucose were lower compared with the HFD group and the same group before treatment (Table 2 and Figure 3). Similar effects were observed on the biochemical profile of the rats with fenofibrate and pioglitazone. Regarding cholesterol concentration, no significant differences were identified among the five groups (Table 2).

On the other hand, there was a reduced level of adiponectin in HFD rats compared to the SD group (Figure 3). As expected, the HFD group had higher insulin secretion and presented insulin resistance compared to the SD. Furthermore, after eight weeks of treatment with ω-3 PUFA, no difference was observed in the secretion of adiponectin and insulin; however, HOMA-IR decreased, and QUICKI increased (Figure 4). This effect was also seen in the pioglitazone group. In the rats with fenofibrate, no differences were observed in adiponectin; however, insulin secretion and HOMA-IR decreased, and QUICKI increased.

#### 2.2.3. Glucose (GTT) and Insulin Tolerance Tests (ITT)

In the HFD group, blood glucose concentrations at 30, 60, 90, and 120 min were higher compared to the SD group. Furthermore, the HFD feeding markedly increased glucose intolerance (Figure 4). ω-3 PUFA, fenofibrate, and pioglitazone treatments for eight weeks in HFD rats significantly decreased the area under the curve (AUC), indicating their prevention of glucose intolerance.

Regarding insulin sensitivity, we confirmed that HFD feeding induces peripheral insulin resistance in rats, as judged by the intraperitoneal insulin tolerance test (Figure 5). ω-3 PUFA treatment increased insulin sensitivity, like fenofibrate and pioglitazone treatments. Finally, the rate constant for glucose disappearance (Kitt) in the HFD group was less than in the SD group. The administration of treatments normalized glucose tolerance to values in SD rats.

#### 2.2.4. Metabolism Modulators in Skeletal Muscle

Some exciting results have been obtained when analyzing the gene and protein expressions of metabolism modulators on skeletal muscle. When studying critical molecules for the insulin signaling pathway, such as INSR, ISR-1, PIK3CA, and GLUT-4, lower expressions have been observed in the group with a HFD compared with the SD rats (Figure 6 and Figure 7). Furthermore, in the group with a HFD, the expressions of the transcription factors PPAR-α and PPAR-γ were significantly lower than the SD group (Figure 8).

After eight weeks of treatment with ω-3 PUFA, the expressions of INSR, ISR-1, PIK3CA, GLUT-4, PPAR-α, and PPAR-γ were higher than that observed in the group with the HFD. Similarly, these effects were also observed in the groups with pioglitazone and fenofibrate. Finally, we verified the expression of these genes by measuring protein expressions in the soleus muscle. As a result, the protein expressions of INSR, ISR-1, PIK3CA, GLUT-4, PPAR-α, and PPAR-γ were higher in the group with ω-3 than that observed in the HFD group (Figure 6, Figure 7 and Figure 8).

## 3. Discussion

The main finding of the present study was that ω-3 PUFA supplementation reverses obesity-associated insulin signaling dysfunction in soleus skeletal muscle. To our knowledge, it is the first time to report that the beneficial effects of ω-3 PUFA are probably associated with the activation of PPARS on INSR, ISR-1, PI3K, and GLUT-4 in the skeletal muscle of male rats fed with a high-fat obesogenic diet.

Our data confirmed that a high-fat diet causes a significant increase in abdominal circumference, Lee index (homolog of BMI), triglycerides, glucose, and body weight. These are characteristic changes of a well-established obesity model [19,20,21]. Furthermore, an increase in the HOMA-IR index and a decrease in QUICKI suggested that HFD rats developed insulin resistance [22,23]. The results obtained from the GTT confirmed an alteration in the metabolism of carbohydrates since the HFD rats had a lower capacity to respond to an oral dose of glucose (glucose concentrations remained high even 120 min after receiving the glucose solution). Moreover, in the ITT, the rate constant for glucose disappearance in obese rats was considerably lower, confirming the decreased sensitivity to insulin [20,24].

After supplementation with ω-3 PUFA, a beneficial effect commonly observed in clinical studies [25,26,27,28,29] and animal models [24,30,31] is the reduction of fasting triglyceride concentration, which is consistent with our results. One of the mechanisms through which omegas ω-3 reduce triglyceride concentration is the stimulation of hepatic β-oxidation through the activation of PPAR-α [32,33]. The activation of PPAR-α plays an essential role in lipid metabolism due to promotes the decreases in the availability of fatty acids for the synthesis of triglycerides; in addition, it is associated with the induction of the activity of lipoprotein lipase (LPL), which is an essential enzyme for the hydrolysis of triglycerides of the VLDL and chylomicrons [32,33]. This background makes us wonder if PPAR-α activation promotes beneficial changes in metabolic modulators in skeletal muscle.

On the other hand, the increase of QUICKI and the decrease of HOMA-IR suggested that supplementation with ω-3 PUFA caused beneficial effects on carbohydrate metabolism and improved insulin sensitivity. Since the HOMA-IR index is an important indicator of insulin resistance. On the contrary, the QUICKI is a good index to estimate insulin sensitivity. Furthermore, we confirmed that ω-3 PUFA has an insulin-sensitizing effect because, in our rats with treatment, the capacity to respond to an oral dose of glucose and the constant rate of glucose disappearance increased. However, the impact of ω-3 PUFAs on glucose concentration and insulin sensitivity has been controversial in previous clinical and preclinical studies. Some clinical studies report that ω-3 PUFA supplementation decreases insulin resistance [34,35]; in contrast, other trials indicate its possible preventive effect [28,36]. In the case of studies implementing animal models, supplementation with ω-3 PUFA appears to prevent and reverse insulin resistance in models of obesity and metabolic syndrome induced by diets high in fat, fructose, or sucrose [31,37,38]. The discrepancy in the effect of ω-3 PUFA in humans could be explained by complications in adherence to treatment [26], genetic background that can generate variability in the response to ω-3 PUFA [39], and the diet consumed by patients due to some studies indicate that nutrients such as fiber, calcium, and magnesium contribute to insulin sensitivity [40].

In the case of anthropometric parameters, our results indicated that ω-3 PUFA supplementation did not change body weight, abdominal circumference, or Lee index. These results were unsurprising because most previous studies reported no changes in these variables [26,40,41]. In cases where ω-3 PUFA supplementation modified anthropometric parameters, the treatment administration period was longer than six months or was complemented with physical activity, dietary, and pharmacological therapy [25,42,43].

By measuring the weight of the soleus skeletal muscle of our rats, it was observed that in our obese group, the absolute weight of the soleus skeletal muscle increased. These results are consistent with what we expected because the soleus is located in the back of the leg, and its function is to maintain and control posture [44]. However, when measuring the relative weight of the soleus, it was observed that it decreased in obese rats. We did not observe the beneficial effect of ω-3 PUFA supplementation on the relative weight of the soleus skeletal muscle.

However, it is necessary to complement the future with more studies about body composition, muscle strength, and physical performance.

On the other hand, due to our data indicating that ω-3 PUFA supplementation has an insulin-sensitizing effect, we determined the gene and protein expression of key components in the insulin signaling pathway in the soleus skeletal muscle. As a result, in our obese rats, a significant decrease in the gene and protein expression of INSR, ISR-1, and PI3K was observed. Consequently, the expression and probably the translocation of GLUT-4 was attenuated. These results suggest a severe alteration of the first steps of insulin signaling and a decrease in glucose uptake in the soleus cells [37,45]. It should be noted that deregulation of lipid metabolism in skeletal muscle is a common feature in pathologies associated with inflammation, such as obesity [45]. Skeletal muscle is an essential organ for glucose and lipid metabolism because it is one of the largest consumers of glucose in the body. Therefore, altered gene and protein expression of metabolism modulators could lead to insulin resistance, muscle lipid accumulation, and loss of muscle mass [8,45].

When analyzing our results, we observed that in the group with ω-3 PUFA, the expression of INSR, ISR-1, PI3K, and GLUT-4 was normalized. These results confirmed that supplementation with ω-3 PUFA is associated with a beneficial effect on modulators of the insulin signaling pathway; however, it is not possible to know if this effect is direct or indirect. Previous studies indicate that the insulin-sensitizing effect in adipose tissue could be due to various mechanisms. For example, ω-3 PUFAs are anti-inflammatory, stimulate fatty acid oxidation, and modulate mitochondrial function [46,47]. Furthermore, studies in macrophages, mature adipocytes, and mice suggest that G protein-coupled receptor 120 (GPR120) is the most important mediator of the effects of ω-3 PUFA [48,49]; however, this receptor is not expressed in muscle. Consequently, the mechanism responsible for the sensitizing effect of insulin on skeletal muscle is poorly understood.

Due to PPARs having been shown to play essential roles in energy metabolism in skeletal muscle, in this study, their probable involvement in the insulin-sensitizing effects of ω-3 PUFAs was explored. Our results showed a decrease in the genetic and protein expression of PPAR-α, which could indicate a reduction in lipid catabolism because this transcription factor in skeletal muscle promotes β-oxidation, decreases the availability of fatty acids for triglyceride synthesis, and induces lipoprotein lipase (LPL) activity [32,33]. Moreover, in our HFD rats, a reduction in the gene and protein expression of PPAR-γ was also observed. This could partially explain the insulin resistance observed in our obese rats because PPAR-γ promotes the expression of INSR, ISR-1, and GLUT-4 [8,45]. Our results are consistent with what was expected because previous studies in rodents with HFD demonstrated that the obesogenic diet significantly alters the expression of these transcription factors [50,51].

It is essential to highlight that the gene and protein expression of PPAR-α and PPAR-γ were increased after administration of ω-3 PUFA, fenofibrate, and pioglitazone in the HFD rats. Therefore, ω-3 PUFA supplementation can activate both PPAR isoforms in a manner comparable to other agonists in clinical use. It is necessary to mention that despite the evidence collected in previous studies indicating the interaction between ω-3 PUFA supplementation and PPAR activity, the participation of these transcription factors as mediators of the modulator of ω-3 PUFA metabolism in vivo remains to be defined. However, it appears that the effects of ω-3 PUFAs on insulin-sensitizing effects could be due, at least partially, to the activation of PPARs [45,52,53,54]. The supplementation of ω-3 PUFA improved insulin resistance and reversed the metabolic abnormalities observed in the soleus skeletal muscle in the obese model.

## 4. Materials and Methods

### 4.1. Animal Model

All experimental protocols of the model were approved by The Children’s Hospital of Mexico Federico Gómez Ethics Committee under the guidelines of the Norma Oficial Mexicana NOM-062-ZOO-1999 on technical specifications for the production, care, and use of laboratory animals. Male Wistar rats at four weeks old (n = 25) were randomly divided into two experimental groups: standard diet (SD, n = 5) or high-fat diet (HFD, n = 20). The groups were housed in acrylic boxes with light/dark cycles of 12 h at 21 to 23 °C (The Children’s Hospital of Mexico Federico Gómez) with free access to water and food.

The SD group was fed with a regular diet (LabDiet 5001) containing 13.6 kcal % fat, 28.9 kcal % protein, and 57.49 kcal % carbohydrate, and the HFD group with a diet to induce obesity (Frogslab) containing 63.4 kcal % fat, 10.2 kcal % protein, and 26.4% kcal % carbohydrate. During the obese model period (16 weeks), the weekly consumption of food, water, and energy was determined. The food was dispensed in stainless steel feeders inside the acrylic boxes, reducing waste. The quantities of food consumed were determined by implementing the formula: Food consumed (g) = Initial weight of food − (Final weight of food + Loss). The amount of food in grams outside the feeder was considered a loss.

At the end of 16 weeks of established obesity, the group HFD was divided into four subgroups: HFD group (n = 5, saline solution 10% tween 20), HFD+ω-3 PUFA group (n = 5, 200 mg/kg EPA: DHA 2:1), HFD+pioglitazone group (n = 5, 30 mg/kg), and HFD+fenofibrate group (n = 5, 100 mg/kg). Treatment was given by gavage once daily for eight weeks using saline solution 10% tween as vehicle.

### 4.2. Zoometric Parameters and Biochemical Profile

During the six months of the animal model, body weight was determined weekly. Furthermore, the distance between the snout and the anus and the abdominal circumference (centimeters) were determined before and after administering the treatment using an anthropometric tape. With this data, the Lee index was calculated, which is a marker of obesity in rats that is homologous to the BMI in humans: ∛body weight (g)/distance between the snout and the anus (cm) [20]. In addition, serum glucose, cholesterol, and triglyceride levels were measured in the five groups of rats (fasting for 10 h) using a blood sample from the lateral tail vein. The biochemical profile was measured using a portable analyzer (Accutrend plus Roche) and reactive strips of the same brand.

### 4.3. Glucose Tolerance Test

A glucose tolerance test (GTT) was performed at the end of 8 weeks of treatment and after fasting for 10 h. Blood samples were obtained from the lateral tail vein (time 0). Furthermore, the rats were given a glucose solution (2 g/kg body weight) via oral gavage. Finally, blood glucose concentrations were measured again 30, 60, 90, and 120 min after glucose administration. The reference values considered were: 0 min 3.89–5.5 mmol/L; 30 min 6.66–9.44 mmol/L; 60 min 5.55–7.77 mmol/L, and 120 min 3.89–6.66 mmol/L [24].

### 4.4. Insulin Tolerance Test

An oral insulin tolerance test (ITT) was performed after fasting for 10 h. Blood samples were collected from the lateral tail vein (time 0). Moreover, the rats were injected intraperitoneally fast-acting regular insulin (0.5 UI/kg). Blood glucose concentrations were measured again 15, 30, 45, and 60 min after insulin administration. The constant for glucose disappearance rate during the test (Kitt) was calculated using the 0.693/t1/2 formula. The glucose t1/2 was calculated from the slope of the least-square analysis of the plasma glucose concentrations during the linear decay phase [21]. The reference values considered were blood glucose < 2.6 mmol/L with a 50% reduction from baseline [21,24].

### 4.5. Euthanasia and Dissection

At the end of the model, the animals (fasting for 10 h) underwent decapitation as a method of euthanasia. The technique was selected considering the guidelines of NOM-062-ZOO-1999, the characteristics of the required samples, and the rats (size and weight). After euthanasia, blood samples were obtained from the carotid vein to carry out the ELISA assay. Moreover, the left and right soleus skeletal muscles were dissected.

The deep muscular plane of the leg was identified, and the soleus skeletal muscle was dissected (following the natural lines), separating it from the underlying tendons to determine absolute weight. Subsequently, the relative weight was calculated with the absolute weight of the soleus muscle (g)/Body weight (g) × 100 [25]. Finally, the soleus skeletal muscle was stored at −30 °C with QIAzol lysis reagent for later processing.

### 4.6. Serum Concentration of Adiponectin, Insulin, and Determination of HOMA-IR and QUICKI

The enzyme-linked immunosorbent assay (ELISA) was implemented to determine insulin and adiponectin concentrations. For this purpose, the blood sample obtained during euthanasia was centrifuged at 900× *g* for 15 min at four °C. The manufacturer’s instructions were followed for using the kits (Millipore: Cat. # EZRMI-13K, Cat. # EZRADP-62K). Furthermore, was estimated the homeostasis model assessment insulin resistance (HOMA-IR): fasting glucose (mg/dL) × fasting insulin (mU/mI)/405) [23]. Finally, we calculated the quantitative insulin sensitivity check index (QUICKI): 1/(log fasting insulin (mU/mL) + log fasting glucose (mg/dL) [23].

### 4.7. Real-Time Polymerase Chain Reaction (qPCR)

Total RNA was extracted from soleus skeletal muscle tissue (40–50 mg) using QIAzol lysis reagent (Qiagen). Total RNA (2 μg) was reverse transcribed into cDNA using the IMPROM II kit (Promega). The quantitative polymerase chain reaction amplifications were quantified using a Light Cycler 480 SYBR Green I Master (Roche). The genes analyzed were INSR, IRS-1, PI3K-CA, GLUT-4, PPAR-α, and PPAR-γ, with GAPDH as a reference gene. The primers used are shown in Table 3.

### 4.8. Western Blotting

Proteins were harvested from the rat’s soleus skeletal muscle (50 mg) using cell lysis buffer (10 mM HEPES at pH 7.9, 10 mM KCI, 1.5 mM MgCl_2_, and 1 mM dithiothreitol). The total proteins (45 µg/mL) were separated in 10% polyacrylamide gels with SDS and a PVDF membrane (Bio-Rad) and transferred into a Transblot Turbo transference system (Bio-Rad). The membranes were preincubated for 1.5 h at ambient temperature with 8% nonfat milk in tris buffered solution (100 mM NaCl, 20 mM Tris; pH 7.6) supplemented with 0.05% Tween 20. Next, they were incubated with specific primary antibodies (INSR: sc-09 Santa Cruz Biotechnology 1:500; Phospho-IRS1 (Tyr612): 44-816G Invitrogen 1:500; PI3K-C2α: sc-365290 Santa Cruz Biotechnology 1:500; PPAR-γ: sc-7273 Santa Cruz Biotechnology 1:500; PPAR-α: PA1-822A Invitrogen 1:500; GLUT-4: G4048 Sigma-Aldrich 1:500 and, β-actin: sc-47778 Santa Cruz Biotechnology 1:500) overnight at four °C and then with a horseradish peroxidase-conjugated secondary antibody (Goat Anti-Rabbit IgG H+L: 111-035-003 Jackson ImmunoResearch 1:1000; Goat Anti-Mouse IgG H+L: 115-035-003 Jackson ImmunoResearch 1:1000) at ambient temperature. Bands were visualized with enhanced chemiluminescence reagents (Bio-Rad, Hercules, CA, USA). Band densities were quantified using ImageJ software (version 8, NIH).

### 4.9. Statistical Analysis

All the results were obtained from 3 to 5 independent experiments. The statistical analysis was performed using GraphPad 8.0 Prism (GraphPad Software 8, San Diego, CA, USA). Data were presented as mean ± SEM. Student’s *t*-tests were used to assess between-group differences, and multiple-group comparisons were performed with one-way analysis of variance followed by Tukey Kramer’s post hoc test. The results were considered statistically significant at *p* < 0.05.

## 5. Conclusions

Our present study confirmed that supplementation of ω-3 PUFA improved insulin resistance and reversed the metabolic abnormalities observed in the soleus skeletal muscle in the obese model by activation of fatty acid oxidation modulators and the insulin signaling pathway. Further studies should be carried out to assess the exact mechanism of ω-3 PUFA on the activation of PPARs and insulin signaling pathways.

## Figures and Tables

**Figure 1 pharmaceuticals-17-00222-f001:**
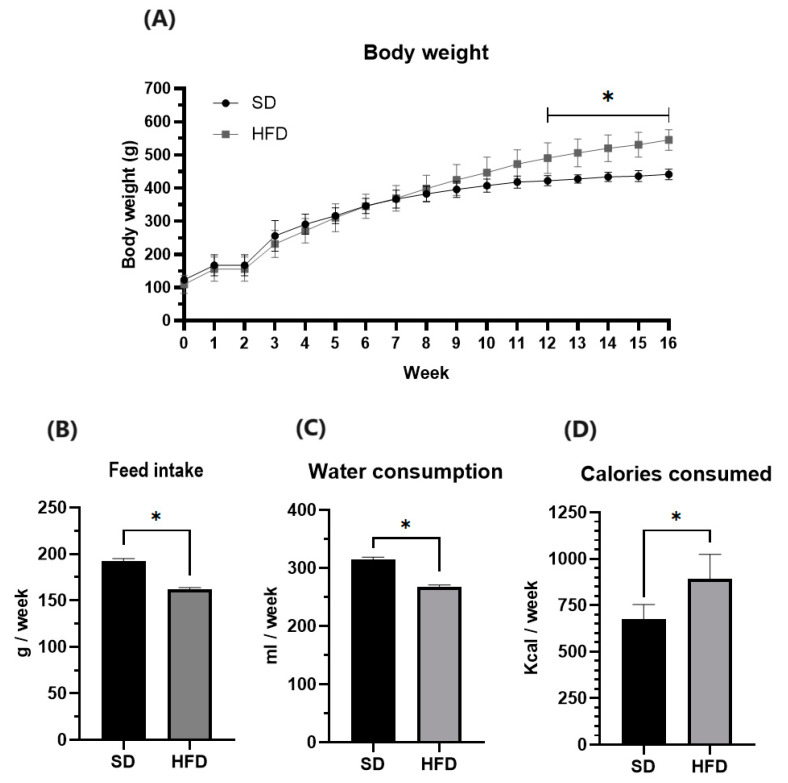
Effect of HFD in (**A**): body weight (g), (**B**): food intake (g/week), (**C**): water consumption (ml/week), and (**D**): calories consumed (kcal/week) of male Wistar rats during the obesity induction. Data are mean ± standard error of the mean. * Significantly different from the SD group (*p* < 0.05). SD, standard diet (n = 5); HFD, high-fat diet (n = 20).

**Figure 2 pharmaceuticals-17-00222-f002:**
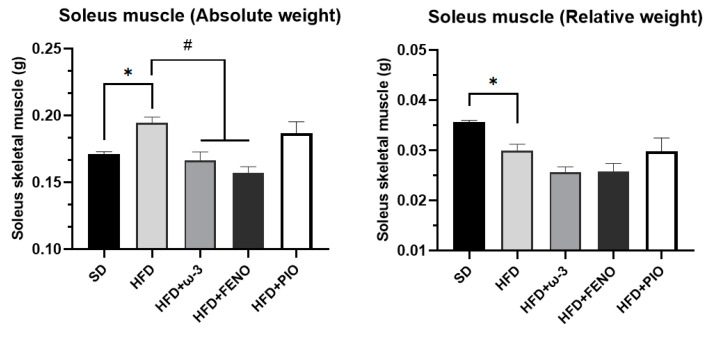
Effect of ω-3 PUFA in the absolute and relative weights of soleus skeletal muscle (g) of male Wistar rats treated for eight weeks. Data are mean ± standard error of the mean. * Significantly different from the SD group (*p* < 0.05). # Significantly different from the HFD group (*p* < 0.05). SD, rats fed the standard diet (n = 5); HFD, rats fed the high-fat diet (n = 5); HFD + ω-3 PUFA, rats fed the high-fat diet with EPA: DHA supplementation (2:1) 200 mg/kg (n = 5); HFD + FENO, rats fed the high-fat diet treated with fenofibrate 100 mg/kg (n = 5); HFD + PIO, rats fed the high-fat diet treated with pioglitazone 30 mg/kg (n = 5).

**Figure 3 pharmaceuticals-17-00222-f003:**
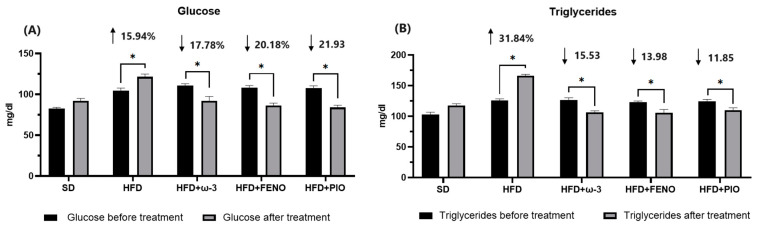
Effect of ω-3 PUFA in (**A**): glucose concentration (mg/dL), (**B**): triglyceride concentration (mg/dL), of male Wistar rats. Data are mean ± standard error of the mean. * Significantly different from the group before treatment (*p* < 0.05). SD, rats fed the standard diet (n = 5); HFD, rats fed the high-fat diet (n = 5); HFD + ω-3 PUFA, rats fed the high-fat diet with EPA: DHA supplementation (2:1) 200 mg/kg (n = 5); HFD + FENO, rats fed the high-fat diet treated with fenofibrate 100 mg/kg (n = 5); HFD + PIO, rats fed the high-fat diet treated with pioglitazone 30 mg/kg (n = 5).

**Figure 4 pharmaceuticals-17-00222-f004:**
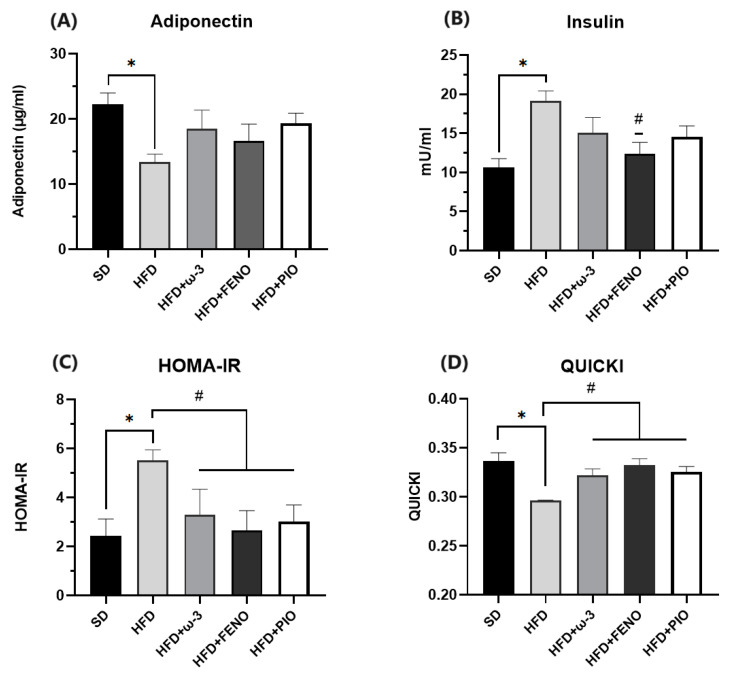
Effect of ω-3 PUFA in (**A**): plasma adiponectin concentration (µg/mL), (**B**): plasma insulin concentration (mU/mL), (**C**): HOMA-IR and (**D**): QUICKI of male Wistar rats. Data are mean ± standard error of the mean. * Significantly different from the SD group (*p* < 0.05). # Significantly different from the HFD group (*p* < 0.05). SD, rats fed the standard diet (n = 5); HFD, rats fed the high-fat diet (n = 5); HFD + ω-3 PUFA, rats fed the high-fat diet with EPA: DHA supplementation (2:1) 200 mg/kg (n = 5); HFD + FENO, rats fed the high-fat diet treated with fenofibrate 100 mg/kg (n = 5); HFD + PIO, rats fed the high-fat diet treated with pioglitazone 30 mg/kg (n = 5).

**Figure 5 pharmaceuticals-17-00222-f005:**
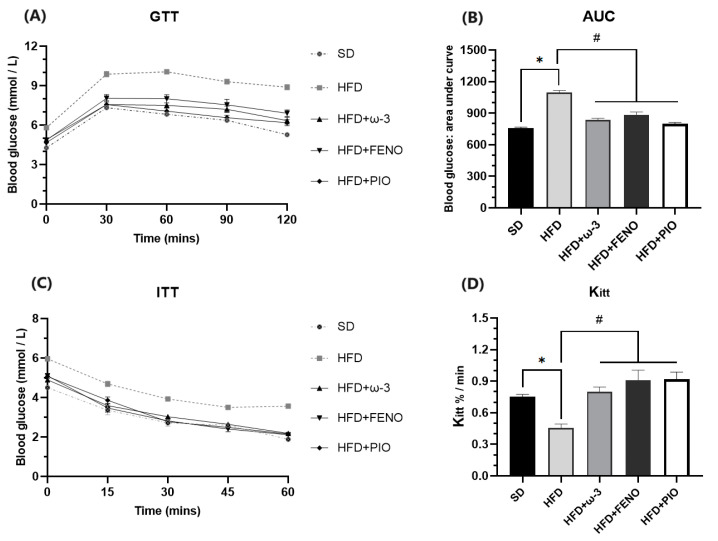
Effect of ω-3 PUFA in (**A**): glucose tolerance tests (GTT), (**B**): area under the curve (AUC), (**C**): insulin tolerance tests (ITT), and (**D**): rate of blood glucose disappearance (Kitt) values of male Wistar rats. Data are mean ± standard error of the mean. * Significantly different from the SD group (*p* < 0.05). # Significantly different from the HFD group (*p* < 0.05). SD, rats fed the standard diet (n = 5); HFD, rats fed the high-fat diet (n = 5); HFD + ω-3 PUFA, rats fed the high-fat diet with EPA: DHA supplementation (2:1) 200 mg/kg (n = 5); HFD + FENO, rats fed the high-fat diet treated with fenofibrate 100 mg/kg (n = 5); HFD + PIO, rats fed the high-fat diet treated with pioglitazone 30 mg/kg (n = 5).

**Figure 6 pharmaceuticals-17-00222-f006:**
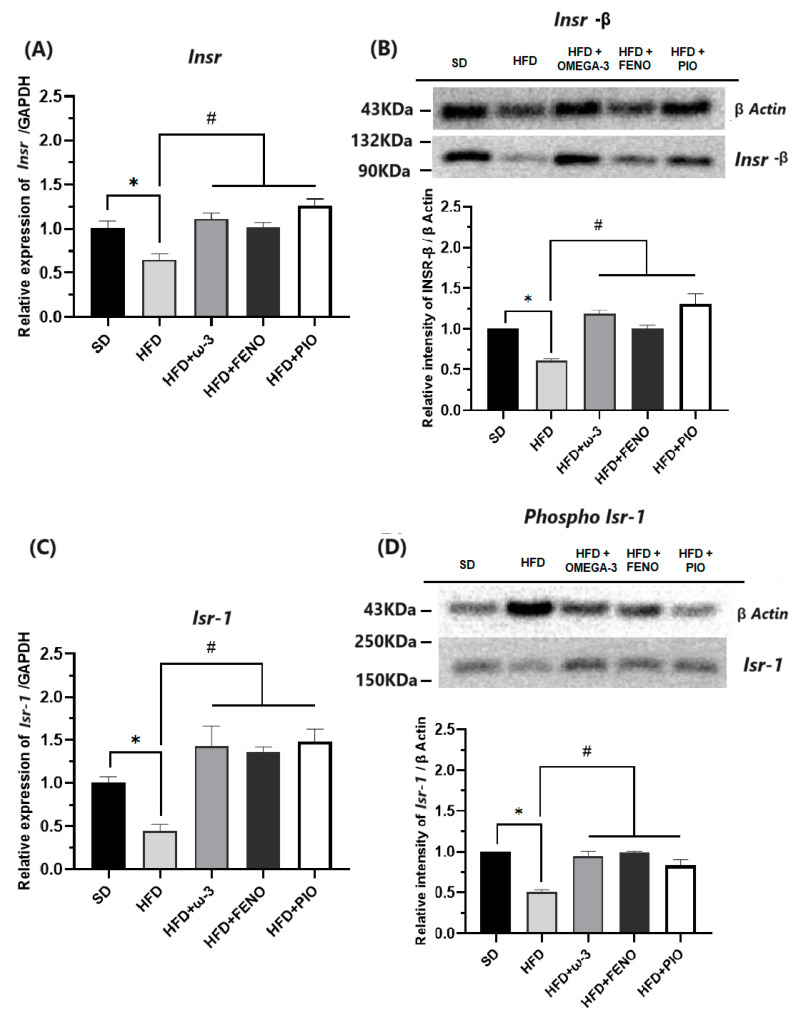
Effect of ω-3 PUFA in (**A**): relative expression of INSR, (**B**): relative intensity of INSR, (**C**): relative expression of ISR-1, and (**D**): relative intensity of Phospho-ISR1 in the muscle in male Wistar rats after 8 weeks of treatment. * Significantly different from the group before treatment (*p* < 0.05). # Significantly different from the HFD group (*p* < 0.05). SD, rats fed the standard diet (n = 5); HFD, rats fed the high-fat diet (n = 5); HFD + ω-3 PUFA, rats fed the high-fat diet with EPA: DHA supplementation (2:1) 200 mg/kg (n = 5); HFD + FENO, rats fed the high-fat diet treated with fenofibrate 100 mg/kg (n = 5); HFD + PIO, rats fed the high-fat diet treated with pioglitazone 30 mg/kg (n = 5).

**Figure 7 pharmaceuticals-17-00222-f007:**
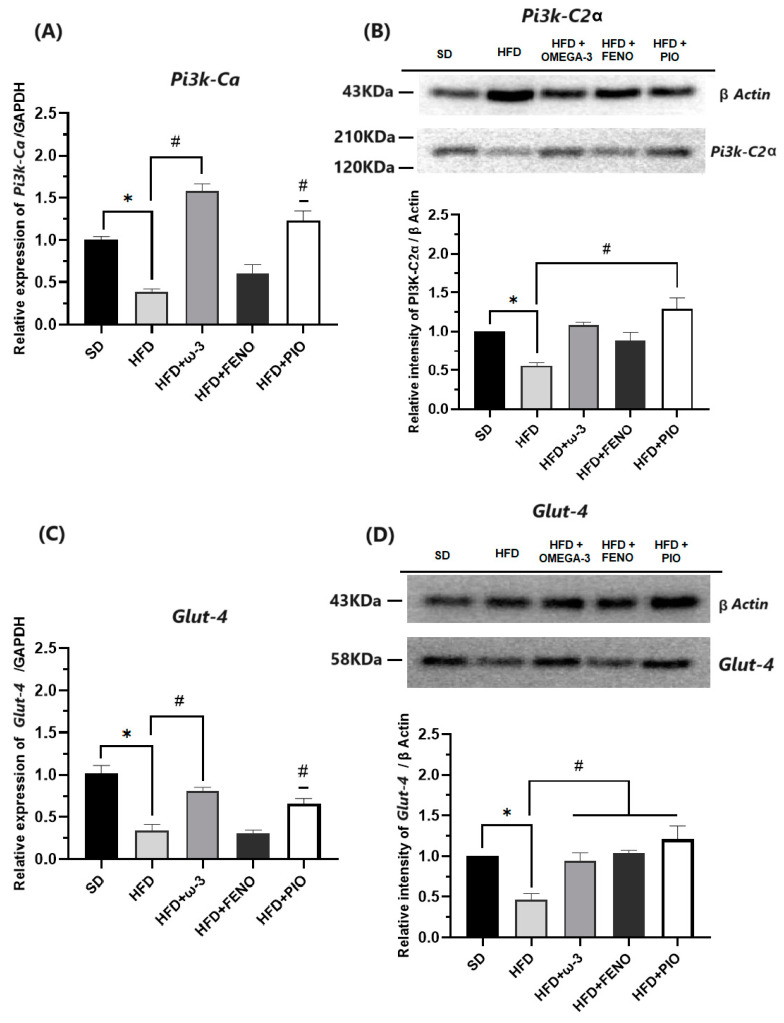
Effect of ω-3 PUFA in (**A**): relative expression of PI3K-CA, (**B**): relative intensity of PI3K-C2α, (**C**): relative expression of GLUT-4, and (**D**): relative intensity of GLUT-4 in the muscle in male Wistar rats after 8 weeks of treatment. * Significantly different from the group before treatment (*p* < 0.05). # Significantly different from the HFD group (*p* < 0.05). SD, rats fed the standard diet (n = 5); HFD, rats fed the high-fat diet (n = 5); HFD + ω-3 PUFA, rats fed the high-fat diet with EPA: DHA supplementation (2:1) 200 mg/kg (n = 5); HFD + FENO, rats fed the high-fat diet treated with fenofibrate 100 mg/kg (n = 5); HFD + PIO, rats fed the high-fat diet treated with pioglitazone 30 mg/kg (n = 5).

**Figure 8 pharmaceuticals-17-00222-f008:**
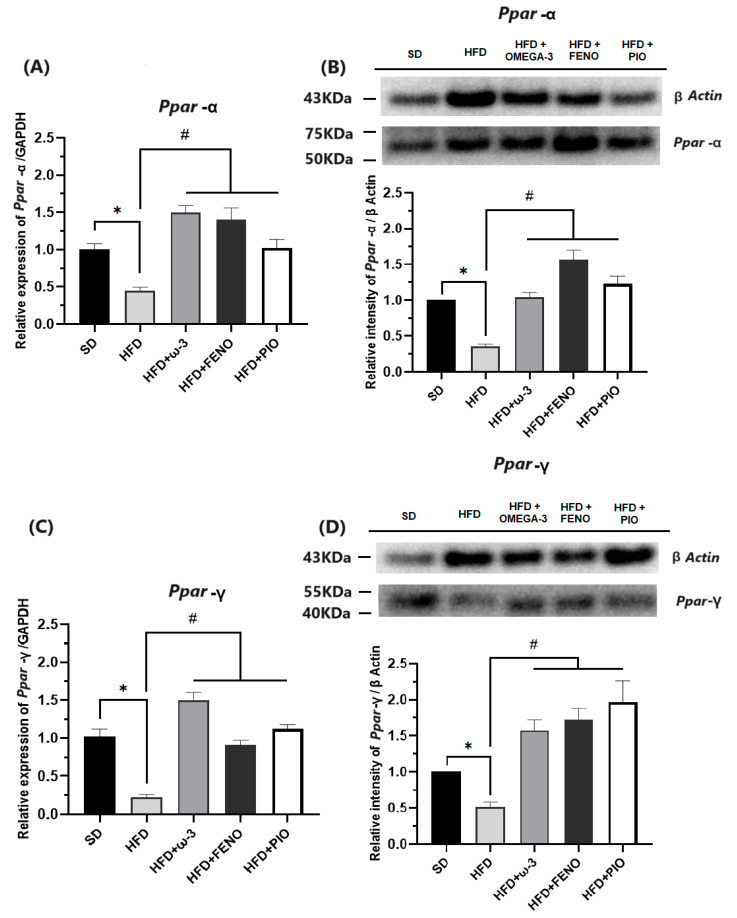
Effect of ω-3 PUFA in (**A**): relative expression of PPAR-α, (**B**): relative intensity of PPAR-α, (**C**): relative expression PPAR-γ and (**D**): relative intensity of PPAR-γ in the muscle in male Wistar rats after 8 weeks of treatment. * Significantly different from the group before treatment (*p* < 0.05). # Significantly different from the HFD group (*p* < 0.05). SD, rats fed the standard diet (n = 5); HFD, rats fed the high-fat diet (n = 5); HFD + ω-3 PUFA, rats fed the high-fat diet with EPA: DHA supplementation (2:1) 200 mg/kg (n = 5); HFD + FENO, rats fed the high-fat diet treated with fenofibrate 100 mg/kg (n = 5); HFD + PIO, rats fed the high-fat diet treated with pioglitazone 30 mg/kg (n = 5).

**Table 1 pharmaceuticals-17-00222-t001:** Effect of HFD in zoometric parameters and biochemical profile of male Wistar rats.

	SD	HFD
Initial body weight (g)	123.1 ± 2.81	109.1 ± 6.06
Final body weight (g)	441.5 ± 7.26	545.1 ± 6.98 *
Abdominal circumference (cm)	19.0 ± 0.17	21.0 ± 0.13 *
Index lee	0.356 ± 0.001	0.366 ± 0.0008 *
Triglycerides (mg/dL)	105.6 ± 2.50	124.9 ± 1.40 *
Glucose (mg/dL)	82.4 ± 1.63	107.7 ± 1.38 *
Cholesterol (mg/dL)	158.8 ± 2.10	164.5 ± 0.82

Data are mean ± standard error of the mean. * Significantly different from the SD group (*p* < 0.05). SD, standard diet (n = 5); HFD, high-fat diet (n = 20).

**Table 2 pharmaceuticals-17-00222-t002:** Effect of ω-3 PUFA supplementation in zoometric parameters and biochemical profile of male Wistar rats.

	SD	HFD	HFD + ω-3PUFA	HFD + FENO	HFD + PIO
Body weight (g)	479.8 ± 7.2	649.3 ± 29.73 *	651.80 ± 17.12	617.1 ± 30.05	636.2 ± 27.90
Abdominal circumference (cm)	19.5 ± 0.17	22.8 ± 0.66 *	22.1 ± 0.18	21.5 ± 0.17	22.2 ± 0.22
Index lee	0.359 ± 0.0008	0.373 ± 0.0008 *	0.370 ± 0.0004	0.369 ± 0.0004	0.373 ± 0.0013
Triglycerides (mg/dL)	117.6 ± 2.90	166.0 ± 2.59 *	106.6 ± 2.23 #	105.8 ± 5.36 #	110.0 ± 3.84 #
Glucose (mg/dL)	92.0 ± 3.08	121.60 ± 3.23 *	92.00 ± 5.35 #	86.2 ± 3.08 #	84.0 ± 2.70 #
Cholesterol (mg/dL)	163.6 ± 2.01	166.8 ± 1.15	161.4 ± 1.85	159.2 ± 2.41 #	166.0 ± 1.11

Data are mean ± standard error of the mean. * Significantly different from the SD group (*p* < 0.05). # Significantly different from the HFD group (*p* < 0.05). SD, rats fed the standard diet (n = 5); HFD, rats fed the high-fat diet (n = 5); HFD + ω-3 PUFA, rats fed the high-fat diet with EPA: DHA supplementation (2:1) 200 mg/kg (n = 5); HFD + FENO, rats fed the high-fat diet treated with fenofibrate 100 mg/kg (n = 5); HFD + PIO, rats fed the high-fat diet treated with pioglitazone 30 mg/kg (n = 5).

**Table 3 pharmaceuticals-17-00222-t003:** The gene sequences used as forward (F) and reverse (R) primers for real-time qPCR.

Gene	Primer Sequence (5′–3′)	Gene Bank
*Insr*	F: TCAGAACCCGATGACCCTAC R: GGGATGCACTTGTTGTTGTG	NM_017071.2
*Irs-1*	F: GCTCTAGTGCTTCCGTGTCC R: GTTGCCACCCCTAGACAAAA	NM_012969.2
*Pi3k-Ca*	F: CATCAGTGGCTCAAGGACAA R: CAGCTGTCCGTCATCTTTCA	NM_133399.3
*Glut-4*	F: CCTCCAGGATGAAGGAAACA R: GGGTAAGAGGAAGGCAGGAC	NM_012751.1
*Ppar-* *α* *,*	F: CTCGTGCAGGTCATCAAGAA R: CAGCCCTCTTCATCTCCAAG	NM_013196.2
*Ppar-* *γ*	F: CTGGCCTCCCTGATGAATAA R: GGCGGTCTCCACTGAGAATA	NM_001145366.1
*Gapdh*	F: AGACAGCCGCATCTTCTTGT R: TTCCCATTC TCAGCCTTGAC	NM_017008.4

## Data Availability

Data is contained within the article.

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
