# Peer review of "Effect of Supplementation with Omega-3 Polyunsaturated Fatty Acids on Metabolic Modulators in Skeletal Muscle of Rats with an Obesogenic High-Fat Diet"

_pharmaceuticals, 2024, doi:10.3390/ph17020222_

Round 1

Reviewer 1 Report

Comments and Suggestions for Authors

Authors should focus on explaining their findings in detail in the discussion section and place them in a context that links them to the aims of the study and the wider field. They should talk about the implications of their findings in a way that is understandable to a wider audience and address possible limitations more directly.

Also, authors should simplify long sentences and give short, simple explanations of results throughout the article to improve readability.

Author Response

Comments 1

Authors should focus on explaining their findings in detail in the discussion section and place them in a context that links them to the aims of the study and the wider field.

Response and Revisions

We reviewed and edited the discussion of results to explain our findings in greater detail and clarity (page 10).

 Comments 2

They should talk about the implications of their findings in a way that is understandable to a wider audience and address possible limitations more directly.
Response and Revisions

The effects of ω-3 PUFAs on insulin-sensitizing effects could be due, at least partially, to activation of PPARs. The supplementation of ω-3 PUFA improved insulin resistance and reversed the metabolic abnormalities observed in the soleus skeletal muscle in the obese mode (Discussion, Pages 11).

 Comments 3

Also, authors should simplify long sentences and give short, simple explanations of results throughout the article to improve readability.

Response and Revisions

To facilitate reading the article, the sentences were shortened, and the ideas were simplified, especially in the introduction and discussion of results.

Reviewer 2 Report

Comments and Suggestions for Authors

Review comments

The manuscript mainly explores the effects of omega-3 polyunsaturated fatty acids on metabolic modulators in the skeletal muscle of rats supplemented with a high-fat diet.

The manuscript requires revisions.

Other comments

1.   Abstract

a.      Give values.

b.     Give online on future direction.

2.   Introduction

a.      Have previous studies used PUFA supplementation?

b.     The authors should clearly highlight the research gap and aim of the study.

3.   Results

a.      Give the percentage of weight increase.

b.     Emphasis on the relationship of obesity to the gut microbiota.

4.   Discussion

a.      The conclusion should be short and precise.

5.   Materials & Methods

a.      How did the authors make sure that food was not wasted?

b.     Which group was given the standard diet?

c.      How did the authors separate the soleus muscle?

d.     When was the GTT carried out? Is it after 16 weeks after treatments?

e.      Give references for GTT.

f.      The same goes for ITT.

6.   References

a.      Websites – give access dates.

Author Response

Abstract

Give values.

Response and Revisions

In the article (summary, Line 11, Page 1), the percentages of increase or decrease in the biochemical profile, HOMA-IR and QUICKI, were added to emphasize the observed changes associated with ω-3 PUFA supplementation.

 Give online on future direction.

Response and Revisions

We add a brief sentence that allows us to conclude our summary better and propose a future perspective (summary, Line 15, Page 1).

 Introduction

Have previous studies used PUFA supplementation?

Response and Revisions

Yes, previous studies provide evidence of the benefits of ω-3 PUFA on the heart, blood vessels, liver, immune system, and endocrine system. We include more information to complement the introduction (introduction, Line .28, Page 2).

The authors should clearly highlight the research gap and aim of the study.

Response and Revisions

We rewrote our objectives in the introduction of this study to better explain its scope (introduction, Line 37, Page 2). The objective of the present study was to evaluate whether ω-3 PUFA supplementation reverts alterations in metabolic modulators associated with a high-fat obesogenic diet in skeletal muscle.

Results

Give the percentage of weight increase.

Response and Revisions

After 16 weeks of the HFD diet, the animal model of obesity was confirmed with 23.46% weight gain compared to the SD group (results, Line 1, Page 2).

Emphasis on the relationship of obesity to the gut microbiota.

Response and Revisions

We do not understand why the intestinal microbiota has an essential role in the subject of the present study. Could you provide us with more information to respond to your request?

Discussion

The conclusion should be short and precise.

Response and Revisions

We reviewed and edited the discussion of results to explain our findings in greater detail and clarity (page 10).

Materials & Methods

How did the authors make sure that food was not wasted?

Response and Revisions

Two different diets were used to feed the groups of rats: standard diet (SD) and high-fat diet (HFD). Both groups of rats had free access to food and water. The food was dispensed in stainless steel feeders inside the acrylic boxes, reducing waste. The quantities of food consumed were determined daily at 2:00 p.m. ± 10 minutes, CDMX time, implementing the formula: Food consumed (g) = Initial weight of food – (Final weight of food + Loss). The amount of food in grams outside the feeder was considered a loss (Materials & methods - Animal model, Line 12, Page 12).

Which group was given the standard diet?

Response and Revisions

Male Wistar rats at four weeks old (n=25) were randomly divided into two experimental groups: standard diet (SD, n=5) or high-fat diet (HFD, n=20) (materials & methods - Animal model, Line 6, Page 11).

 How did the authors separate the soleus muscle?

Response and Revisions

The deep muscular plane of the leg was identified, and the soleus skeletal muscle was dissected (following the natural lines), separating it from the underlying tendons (Materials & methods - Animal model, Line 1, Page 12).

 When was the GTT carried out? Is it after 16 weeks after treatments?

Response and Revisions

A glucose tolerance test (GTT) was performed at the end of 8 weeks of treatment and after fasting for 10 hours (Materials & methods - Glucose tolerance test, Line 1, Page 12)

 Give references for GTT.

Response and Revisions

The reference values considered were: 0 minutes 3.89-5.5 mmol/L; 30 minutes 6.66-9.44 mmol/L; 60 minutes 5.55-7.77 mmol/L and 120 minutes 3.89-6.66 mmol/L (Materials & methods - Glucose tolerance test, Line 5, Page 12).

The same goes for ITT.

Response and Revisions

Blood glucose < 2.6 mmol/L with a 50% reduction from baseline (Materials & methods - Insulin tolerance test, Line 7, Page 12).

References

Websites – give access dates.

Response and Revisions

We added the requested information to complete the website references (references 2 and 3, Page 15).

World Health Organization. 2020. Obesity and overweight. http://origin.who.int (July 15, 2023).

World Obesity Federation. World Obesity Atlas. 2023. https://www.worldobesity.org/ (August 7, 2023)

Reviewer 3 Report

Comments and Suggestions for Authors

Comments and Suggestions

1.      I will suggest rewriting the conclusion in a better and simple way.

2.      While the introduction highlights the lack of exploration regarding the impact of ω-3 PUFA supplementation on metabolic modulators in skeletal muscle, it could be improved by briefly summarizing existing research on this to further emphasize the research gap.

3.      Describe the method of gavage in more detail. Include information about the frequency, duration, and any precautions taken during administration.

4.      Include information about how the glucose and insulin tolerance tests were performed. Mention any fasting duration and conditions.

5.      How much total proteins were used for the western blot

6.      I will suggest adding dilutions of all primary and secondary antibodies in the manuscript.

7.      I will suggest adding the specific molecular weights of the proteins in Western blot images.

8.      Clarify why decapitation was chosen as the method of euthanasia and how it complies with ethical guidelines.

9.      When discussing the effect of omega-3 PUFA supplementation, the results indicate that there was no significant difference in body weight, abdominal circumference, and Lee index between the obese rats with ω-3 PUFA and the HFD group. The lack of significant differences should be emphasized, as this is a key finding.

10.   The manuscript discusses the changes in triglyceride and glucose levels, but the relative percentages of change should be provided to highlight the magnitude of the differences between groups.

11.   The explanation of the HOMA-IR and QUICKI results should be more detailed. What do these indices signify, and why is it significant that ω-3 PUFA supplementation affects them?

12.   The mechanisms underlying the effects of ω-3 PUFA on triglyceride concentration and insulin sensitivity are discussed but could be more clearly explained. The discussion could delve into the molecular pathways involved, providing a deeper understanding of how ω-3 PUFA exerts its effects.

13.   The effects of ω-3 PUFA on skeletal muscle are mentioned briefly, but the significance of these effects and their potential implications for muscle health and function could be elaborated upon.

Author Response

Comments 1

 I will suggest rewriting the conclusion in a better and simple way.

Response and Revisions

We reviewed and edited the discussion of results to explain our findings in greater detail and clarity (page 10).

 Comments 2

While the introduction highlights the lack of exploration regarding the impact of ω-3 PUFA supplementation on metabolic modulators in skeletal muscle, it could be improved by briefly summarizing existing research on this to further emphasize the research gap.

Response and Revisions

We include more information to complement the introduction (introduction, Line .28, Page 2). We briefly indicate that previous studies provide evidence of the benefits of ω-3 PUFA on the heart, blood vessels, liver, immune system, and endocrine system.

 Comments 3

Describe the method of gavage in more detail. Include information about the frequency, duration, and any precautions taken during administration.

Response and Revisions

Two different diets were used to feed the groups of rats: standard diet (SD) and high-fat diet (HFD). Both groups of rats had free access to food and water. During the obese model period (16 weeks), the weekly consumption of food, water, and energy was determined. The food was dispensed in stainless steel feeders inside the acrylic boxes, reducing waste. The quantities of food consumed were determined daily at 2:00 p.m. ± 10 minutes, CDMX time, implementing the formula: Food consumed (g) = Initial weight of food – (Final weight of food + Loss). The amount of food in grams outside the feeder was considered a loss (Materials & methods - Animal model, Line 11, Page 12).

 Comments 4

Include information about how the glucose and insulin tolerance tests were performed. Mention any fasting duration and conditions.

Response and Revisions

A glucose tolerance test (GTT) was performed at the end of 8 weeks of treatment and after fasting for 10 hours. Blood samples were obtained from the lateral tail vein (time 0). Furthermore, the rats were given a glucose solution (2 g/kg body weight) via oral gavage. Finally, blood glucose concentrations were measured again 30, 60, 90, and 120 minutes after glucose administration. The reference values considered were: 0 minutes 3.89-5.5 mmol/L; 30 minutes 6.66-9.44 mmol/L; 60 minutes 5.55-7.77 mmol/L and 120 minutes 3.89-6.66 mmol/L (Materials & methods - Glucose tolerance test, Line 1, Page 12).

 Comments 5

How much total proteins were used for the western blot?

Response and Revisions

The total proteins (45 µg/ml) were separated in 10% polyacrylamide gels with SDS and a PVDF membrane (Bio-Rad) and transferred into a Transblot Turbo transference system (Bio-Rad) (Materials & methods - Western Blotting, Line 3, Page 13).

 Comments 6

I will suggest adding dilutions of all primary and secondary antibodies in the manuscript.

Response and Revisions

Next, they were incubated with specific primary antibodies (INSR: sc-09 Santa Cruz Biotechnology 1:500; Phospho-IRS1 (Tyr612): 44-816G Invitrogen 1:500; PI3K-C2α: sc-365290 Santa Cruz Biotechnology 1:500; PPAR-γ: sc-7273 Santa Cruz Biotechnology 1:500; PPAR-α: PA1-822A Invitrogen 1:500; GLUT-4: G4048 Sigma-Aldrich 1:500 and, β-actin: sc-47778 Santa Cruz Biotechnology 1:500) overnight at four °C and then with a horseradish peroxidase-conjugated secondary antibody (Goat Anti-Rabbit IgG H+L: 111-035-003 Jackson ImmunoResearch 1:1000; Goat Anti-Mouse IgG H+L: 115-035-003 Jackson ImmunoResearch 1:1000) at ambient temperature (Materials & methods - Western Blotting, Line 7, Page 13).

 Comments 7

I will suggest adding the specific molecular weights of the proteins in Western blot images.

Response and Revisions

We added the requested information (Figures 6, 7, and 8, Pages 7, 8, and 9).

 Comments 8

Clarify why decapitation was chosen as the method of euthanasia and how it complies with ethical guidelines.

Response and Revisions

The technique was selected considering the guidelines of NOM-062-ZOO-1999 on technical specifications for the production, care, and use of laboratory animals. Furthermore, it's necessary to consider the characteristics of the required samples and the rats (size and weight).

Because the rats implemented in this study are obese, their size does not allow the use of cervical dislocation as a method of euthanasia. Moreover, we require tissues and blood free of chemical contamination (modifies metabolic activity) and cannot use sodium pentobarbital. Therefore, the most appropriate mechanism allowed by the NOM-062-ZOO-1999 is decapitation.

Comments 9

When discussing the effect of omega-3 PUFA supplementation, the results indicate that there was no significant difference in body weight, abdominal circumference, and Lee index between the obese rats with ω-3 PUFA and the HFD group. The lack of significant differences should be emphasized, as this is a key finding.

Response and Revisions

In the case of anthropometric parameters, our results indicated that ω-3 PUFA supplementation did not change body weight, abdominal circumference, or Lee index. These results were unsurprising because most previous studies reported no changes in these variables. In cases where ω-3 PUFA supplementation modified anthropometric parameters, the treatment administration period was longer than six months or was complemented with physical activity, dietary, and pharmacological therapy (Discussion, Line 41, Pages 10).

 Comments 10

The manuscript discusses the changes in triglyceride and glucose levels, but the relative percentages of change should be provided to highlight the magnitude of the differences between groups.

Response and Revisions

We provide the percentages of change to highlight the magnitude of the effect of ω-3 PUFA supplementation on glucose and triglyceride concentrations (Figures 3, Pages 5).

 Comments 11

The explanation of the HOMA-IR and QUICKI results should be more detailed. What do these indices signify, and why is it significant that ω-3 PUFA supplementation affects them?

Response and Revisions

On the other hand, the increase of QUICKI and the decrease of HOMA-IR suggested that supplementation with ω-3 PUFA caused beneficial effects on carbohydrate metabolism and improved insulin sensitivity.  Since the HOMA-IR index is an important indicator of insulin resistance. On the contrary, the QUICKI is a good index to estimate insulin sensitivity (Discussion, Line 25, Pages 10).

 Comments 12

The mechanisms underlying the effects of ω-3 PUFA on triglyceride concentration and insulin sensitivity are discussed but could be more clearly explained. The discussion could delve into the molecular pathways involved, providing a deeper understanding of how ω-3 PUFA exerts its effects.

Response and Revisions

We reviewed and edited the discussion of results to explain our findings in greater detail and clarity. Furthermore, we want to explain better how ω-3 PUFA exerts its effects (page 10).

 Comments 13 

The effects of ω-3 PUFA on skeletal muscle are mentioned briefly, but the significance of these effects and their potential implications for muscle health and function could be elaborated upon.

Response and Revisions

The effects of ω-3 PUFAs on insulin-sensitizing effects could be due, at least partially, to activation of PPARs. The supplementation of ω-3 PUFA improved insulin resistance and reversed the metabolic abnormalities observed in the soleus skeletal muscle in the obese mode (Discussion, Pages 11)

Round 2

Reviewer 3 Report

Comments and Suggestions for Authors

The authors have responded to all my comments.

Author Response

Muchas gracias por tomarse el tiempo de revisar este manuscrito. He incluido las correcciones detalladas en los archivos reenviados.
